# Predicting Network Hardware Faults through Layered Treatment of Alarms Logs

**DOI:** 10.3390/e25060917

**Published:** 2023-06-09

**Authors:** Antonio Massaro, Dimitre Kostadinov, Alonso Silva, Alexander Obeid Guzman, Armen Aghasaryan

**Affiliations:** Nokia Bell Labs, 12 Rue Jean Bart, 91300 Paris, France; dimitre.kostadinov@nokia-bell-labs.com (D.K.); alonso.silva@nokia-bell-labs.com (A.S.); alexander.obeid_guzman@nokia.com (A.O.G.); armen.aghasaryan@nokia-bell-labs.com (A.A.)

**Keywords:** predictive maintenance, network hardware fault prediction, machine learning

## Abstract

Maintaining and managing ever more complex telecommunication networks is an increasingly difficult task, which often challenges the capabilities of human experts. There is a consensus both in academia and in the industry on the need to enhance human capabilities with sophisticated algorithmic tools for decision-making, with the aim of transitioning towards more autonomous, self-optimizing networks. We aimed to contribute to this larger project. We tackled the problem of detecting and predicting the occurrence of faults in hardware components in a radio access network, leveraging the alarm logs produced by the network elements. We defined an end-to-end method for data collection, preparation, labelling, and fault prediction. We proposed a layered approach to fault prediction: we first detected the base station that is going to be faulty and at a second stage, and using a different algorithm, we detected the component of the base station that is going to be faulty. We designed a range of algorithmic solutions and tested them on real data collected from a major telecommunication operator. We concluded that we are able to predict the failure of a network component with satisfying precision and recall.

## 1. Introduction and Our Contribution

The scale and complexity of modern telecommunication networks call for the intelligent support of human experts in management and maintenance operations. In this work, we put forward a method to predict hardware component failures, with the goal of automatizing the troubleshooting procedures that are currently performed by human experts. Predictive maintenance is important from both a cost reduction and a network performance perspective. Replacing in advance fault-risky components allows for efficient management of stock and delivery costs while preventing service degradation. The earlier the fault is detected, the higher the benefit perceived by the operator. In general, the management of complex systems is a challenge because of non-trivial correspondences between the observed behaviors and the underlying states of interest. In the case of fault management, alarms generated by the monitoring system do not always express (current or future) occurrences of faults that require any repair action.

We consider hardware faults in a radio access network composed of around ten thousand base stations (with a dozen of replaceable components in each). The network generates thousands of monitoring events per hour. Each such event does not necessarily reflect an important fault situation: alarms are present in normal operational states, too. In practice, in such a network, only a few hundred faulty physical components are replaced each year. The challenge is therefore to detect and predict the need for the replacement of network components by analyzing large rates of daily alarm flows, where most of the time the network is in a normal operational state.

If we frame our task as a classification problem, we are immediately faced with the well-known challenge of high imbalance between classes (faulty vs normal components), which results in a bias toward the larger class of normal cases. To address this problem, we propose a layered approach to fault prediction, where, as opposed to a direct per-component classification, we first train a model to identify faulty base stations, and only after we select the most likely faulty component on these shortlisted sites. This procedure reduces the data imbalance by at least a factor of ten. Furthermore, it brings another important advantage. The components located on the same site are functionally interdependent and therefore are subject to fault propagation phenomena. A direct classification method based on individual component alarms would miss that local context, whereas a site-level fault identification which uses all the alarms collected on the site provides a way to remedy this loss of information.

Each base station of a radio network can have a different configuration depending on the coverage area, frequency band, radio technologies used, or expected traffic volumes. They can contain a variable number and different types of hardware components. To leverage the site-level data and enable the training of a generic model for the set of different radio sites, we designed a data summarization method that makes abstraction from each site’s specific configuration while maintaining the relevant information for fault prediction. In its simplest variant, it extracts features in terms of alarm and resource types, however, this summarization method is empowered to build graph summaries representing typed time-space relationships in the original data such as correlations, and topological or causal dependencies.

While most of the general-purpose data-driven ML approaches take for granted the availability of good quality data and ground truth labels, in practice this is a major obstacle to the large-scale industrial deployment of AI. In this study, we focused on the automation of the end-to-end system including the data collection, preparation, labeling, and fault prediction components.

Finally, to evaluate the accuracy of our failure prediction model, we used precision and recall, which are widely used measures in classification problems. Precision refers to the fraction of correctly predicted cases (true positives) among the predicted ones, while recall is the fraction of correctly predicted cases among all the failure cases. As already noted, we are dealing with imbalanced classes; the ratio between normal and faulty sites will still be significant even after the reduction procured by the layered approach. Therefore, it is difficult to achieve both high precision and high recall. However, for practical reasons, achieving high precision is much more important as it will limit additional costs related to false positives (e.g., the costs related to useless on-site human interventions), while reasonably low recall would still allow the operator to maintain the current expert-based operational modes for the faults which are not discovered by the algorithm while benefiting from the prediction for those that are actually discovered. We showed that the trade-off between precision and recall can be tuned according to the business needs.

The contributions of this study are the following:We designed and developed end-to-end automation of the predictive hardware maintenance process for large telco networks (Figure 1). Namely, we addressed the problem of the seamless collection of raw data in high volumes and their automated consolidation from multiple sources and labeling. We remark that, the best of our knowledge, such a process is not in place in current network monitoring products, hence it is a relevant innovation and it represents the first contribution of this work.We developed a layered fault prediction scheme for fault diagnosis which decomposes the decision-making into two steps: site-level and component-level prediction. This reduces the problem of data imbalancing and allows for the application of ML methods to alarm logs data.We introduced the data summarization procedure to allow generalization across different network contexts and site configurations and enable ML-based modeling and prediction of faults. We developed an ad-hoc graph-matching algorithm achieving good accuracy for component-level fault prediction on top of the site-level outcomes.

This paper is organized as follows. In Section 2 we provide an analysis of the related literature. Section 3 and Section 4 formally define the problem statement and the principles of the layered approach, respectively. Then, in Section 5 we describe the process of data collection, consolidation, and automated labeling which are essential parts of a practical data-driven approach. Section 6 describes different algorithmic approaches both at the site and component levels. The numerical results of the experiments and the performance evaluations are presented in Section 7. Finally, we conclude in Section 9.

## 2. State of the Art

The problem of fault detection, prediction, and root-cause analysis is a long-standing research field. In the following, we list works that we believe have some aspects in common with our research.

The task of predicting faulty sites can be approached in multiple directions: the first direction is to consider it as a classification task in which we label normal and faulty sites. Within this direction, we also consider anomaly detection methods which will be trained on normal sites and will evaluate the deviation from normal behavior. One of our techniques follows this approach (RNN). The second direction is to consider it as a regression task in which we could consider only the sites which have failed or consider the last known functioning time of all the sites (see for example [1,2,3] where the authors propose a model for remaining useful life prediction). In the presence of right-censored data (where the event of interest has only occurred for a subset of the observations), the regression task will usually result in an underestimate (if we only consider the events) or an overestimate (if we only consider the last known functioning times) of the time-to-event (see for example [4]). The third direction is to consider it as a time-to-event prediction task (see for example [5,6,7]). One of our techniques follows this approach (time-to-event analysis). Our technique is different from a standard random survival forest because we take into account the evolution of the sites with two extra parameters, the average rolling window and the risk threshold. Estimating the time to failure is a possible way to tackle fault prediction, and it has been explored by several authors. For example, in [8] a deep learning model for time to failure prediction was put forward. Here, the authors tackled the problem stemming from the fact that offline data on which the models are trained can, in practice, follow a different distribution from on-line data, on which the models would be tested, and they proposed a technique based on a variational autoencoder and a long-short term memory network. In [1], the authors proposed a parallel hybrid neural network with integration of spatial and temporal features for remaining useful life prediction where the spatial features are extracted by a 1-dimensional convolutional neural network and the temporal features are extracted by a bidirectional gated RNN.

In [9], the authors used a neural network to classify alarm patterns of a radio network. The task here is to associate each alarm pattern to its root cause, encoded as a class. Each alarm pattern is represented by a binary vector. The authors test their system in a very simplified setting, where the network is composed of five base stations, a subset of 94 alarms is chosen to encode the state of the network, and the root causes are restricted to single-link failures and are known with certainty. Moreover, by its very definition, the classifier cannot generalize to networks with a different number of base stations and a different number of alarms. In our case, we deal with a network of over ten thousand base stations, and we propose a method that can generalize to unseen network topologies.

In [10] the authors tackled the problem of software components’ fault prediction in a commercial telecommunication system. The authors used a semi-Markov chain model, trained on real data, to estimate the probability of incurring a fault within a given time window, given an encoding of the current state of the system. This implies building an explicit state-space representation which can become a bottleneck in terms of computational complexity when faced with large systems. In fact, the authors tested with a period of one-day data collected on a presumably small network by using a stress generator.

In [11], the authors developed a system to detect and predict failures in an IoT system. First, from historical alarms’ data (a small 24-h period), the authors computed, for each couple of alarms, the conditional probability of the first alarm being on given that the second alarm is on. From this information, a so-called causal dependence graph is inferred. Such a graph should model how faults spread from one device to the other, given the dependence information of their alarms. Next, if a fault on some components is detected, the causal-dependence model is used to forecast how such a fault will propagate to other devices. In our case, simple co-occurrence of alarms could not be used to model causal relationships, as one would easily confuse causation and correlation. Moreover, in this paper, we did not address the problem of fault propagation to different parts of the network, but rather we predicted fault occurrence in the first place.

In [12], the authors tackled the problem of fault detection and localization in large enterprise networks. They modeled the set of the services on which each host depends and the set of the network components as a graph. Here, each node maintains a probability distribution over three possible states, namely ‘up’, ‘down’, and ‘troubled’. The authors define an algorithm to fit such probabilities and quantify how the state of network components propagates to the state of network services, from observed data. In the second stage, observations from the states of the services are used to infer the most likely state of each network component and pin down those that are in ‘down’ or ‘troubled’ states. The construction of the graph is based on the observation of the packets that all hosts send and receive in the network, and the performance indicator is the response time of network services. In our case we do not have access to traffic data: we leverage only alarms’ data, emitted from network components. While in [12], it is possible to know which network component may cause a failure because it is known which network components each network service uses, in our case we cannot, as we do not have data on causal relationships between alarms, therefore this method is not directly applicable to our problem. Finally, the proposed model performs failure detection and root-cause analysis, whereas we aim at doing failure prediction.

In [13], the authors proposed an anomaly detection system for OpenStack. They modeled all processes running within OpenStack as a graph, where each node represents a process and edges represent dependencies between processes. Dependencies are constructed using the information of the communications between processes, each of which is associated with a unique TCP/UDP connection to communicate with other processes, which can be identified. Next, statistics on some KPIs of each process are gathered and used to spot anomalous behaviors of any of the processes. Once a node reporting anomalous behavior is detected, the system uses the dependency graph to retrieve the set of processes affected by the anomaly. In this case, the authors constructed a graph that mimics physical dependencies between the entities they need to monitor and can pin down with certainty an anomaly on any of the nodes. On the other hand, we do not know which alarm or which component is giving rise to the anomaly, but just that the system is experiencing a fault.

In [14], a system to create a network alarm correlation model was designed and used for fault diagnosis. Here, the modeled correlations are meant not to be learned from data but rather specified by domain experts, and it is not discussed how such a system could go beyond fault diagnosis and tackle fault prediction and preventive maintenance.

In [15], the authors proposed a method based on the PC algorithm, to construct causal graphs (as opposed to correlation graphs) representing causal relationships between network events, leveraging network logs data. The computation involves the time series of alarms. The idea of using causal graphs instead of correlation graphs could be an alternative worth exploring as a future direction of our work, as it seems from our earlier experiments (not presented in this paper) that simple correlation data do not improve significantly the prediction performance at the site level.

In [16], the problem of fault localization was tackled by resorting to a Bayesian network that models the causal dependencies between hardware faults as probabilities, and the prior probabilities of hardware faults of network components. However, contrary to our case, it was assumed that such causal and probabilistic structure is given, possibly constructed by expert judgment or by machine learning.

In [17], the authors addressed the fault detection and diagnosis problem in cloud infrastructures. The training data were generated with the help of injected, labeled faults. The approach relies on building two graphs from the monitoring data, a substitution graph and a detection graph. The substitution graph allows grouping together the strongly correlated metrics and events into clusters, the members of the clusters are then considered equivalent and can substitute one another, which allows for reducing the complexity of the model. The detection graph comprises the sequences of events that lead to the faults. The authors propose an approach based on Facebook’s Edge-Rank algorithm to find the key events that contribute to each fault, and to form fault patterns that can be exploited in an online fault detection mode. The paper focuses on fault detection and diagnosis as opposed to prediction. Moreover, it heavily exploits real-valued KPI as opposed to alarms in our case. More importantly, the proposed approach was built on a fine-tuned model for a fixed system infrastructure, but it is not clear how the learned models would be transposed to a different configuration.

In [18], the authors addressed the problem of anomaly detection in virtual network function (VNF) chains. They tracked (Pearson) correlation pattern changes between neighboring nodes as indicators of anomalies. The idea is somewhat similar to the methods we investigate in this paper at the module layer algorithms, however, no generalization or prediction capability is provided by the authors.

Finally, in [19,20], we have two other examples of research works that addressed early fault detection and localization problems based on numerous real-valued KPI measurements by leveraging correlations between these metrics. Although in a different technological context and with different types of data (alarms as opposed to KPIs), we have also considered correlation patterns as an important source of information for tracking the changes in the systems with respect to fault occurrences in modules. In the current study, we brought in addition the dependency graph summarization method allowing us to cope with topology variations across sites. Furthermore, regarding all the works sites above we also brought in the advantages of the layered approach in terms of scalability and improved imbalance factor.

## 3. Problem Statement

We wanted to predict faults in network hardware components (In this paper, we use the terms hardware components and modules interchangeably.) of a given site, given an observation of its alarms. Since it is difficult to establish deterministic relationships between the appearance of alarms and hardware faults, we resort to the language of probability. Let us make these concepts precise: we denote by *A* the set of all alarms emitted by a site, and by *C* the set of its components, and let α and γ be the generic subset of *A* and *C*, respectively. Moreover, each network component can emit a particular set of alarms, that may or may not have alarms in common with other components. We denote the alarms emitted by a set of components γ as α(γ). We remark that some alarms cannot be linked to any specific component, rather, they are linked to the entire site. Let us define, for each subset of observed alarms α, the probability distribution of being faulty over all the subsets of network components as P(γ|α). We remark here that P(γ|α) is the probability of all hardware components contained in γ to be faulty, given the observed alarms α.

Now, assuming we know such probability distribution, we would like to pin down, roughly speaking, for any observed set of alarms α, a set of components $\gamma^{*}$ that is most likely to be faulty.
(1)$$\gamma^{*}\in \underset{\gamma \subset C}{arg max}P\left(\gamma |\alpha \right).$$

## 4. Layered Approach

In the following, P(∅|α) will denote the probability of not having any fault in any component. Moreover, we define the event *‘site C is faulty’* as the event in which site *C* hosts at least one faulty hardware component. By definition the two events *∅* and *site C is faulty* are complementary, hence ∅=
*‘C is not faulty’* and P(∅|α)=1−P(C is faulty|α).

We notice that defining the most likely faulty component as in (Equation 1) can lead to paradoxical results. Let us make this clear: by the basic rules of probability theory, we can write
(2)P(γ|α)=P(γ|C is faulty,α)·P(C is faulty|α)+P(γ|C not faulty,α)·P(C not faulty|α).
Now, we observe that, for γ≠∅, by definition, P(γ|C is not faulty,α)=0, therefore we are left with
P(γ|α)=P(γ|C is faulty,α)·P(C is faulty|α)ifγ≠∅
In the same way, for γ=∅, by definition, P(∅|C is faulty,α)=0 and P(∅|Cisnotfaulty,α) = 1, therefore in this case we are left with
P(∅|α)=P(C is not faulty|α).
This means that γ≠∅⇒p(γ|α)=p(γ|C is faulty,α)P(C is faulty|α) and γ=∅⇒p(∅|α)=P(C is not faulty|α), therefore Equation (Equation 1) can be rephrased as
(3)$$\gamma^{*}=arg max\left\{\begin{array}{c} P(\gamma =\emptyset |\alpha ),\\ {arg max}_{\gamma \subset C,\;\gamma \neq \emptyset }P\left(\gamma |C\;is\;faulty,\alpha \right)P\left(C\;is\;faulty|\alpha \right) \end{array}\right\}.$$
Now, from Equation (Equation 3), it is obvious to see that if P(C is not faulty|α)>P(C is faulty|α), then $\gamma^{*}=\emptyset$, since p(γ|C is faulty,α)≤1. This means that, if the probability of not having any fault is higher than the probability of having at least one fault, we would not declare any component as faulty, as one would expect. However, in the opposite case, in which P(C is faulty|α)>P(C is not faulty|α), we could still choose ∅ as $\gamma^{*}$. In fact, if p(γ|C is faulty,α) is sufficiently small for each γ≠∅, then all the products appearing in the inner arg max at Equation (Equation 3) would be smaller than P(C not faulty|α).

This argument shows that blindly applying (Equation 1) can lead to a paradoxical situation where despite the probability of a site being faulty is higher than the probability of the site not being faulty, we still do not raise any warning on any component. This is one of the reasons why we propose a *layered* fault prediction mechanism as follows. At the site layer, if P(C is not faulty|α)>P(C is faulty|α), then we declare that no component is faulty. At the component layer, the respective algorithm is activated if P(C is not faulty|α)≤P(C is faulty|α), and would choose $\gamma^{*}:\approx {arg max}_{\gamma \subset C}P\left(\gamma |C\;is\;faulty,\alpha \right)$. In this way, we always output an estimate for the most likely faulty module(s), in the case in which a site is more likely than not to contain at least a faulty module.
(4)$$\gamma^{*}:\approx \left\{\begin{array}{cc} \emptyset , & \mathrm{if}P(C\;is\;not\;faulty|\alpha )>P(C\;is\;faulty|\alpha )\\ {arg max}_{\gamma \subset C}P\left(\gamma |C\;is\;faulty,\alpha \right), & otherwise \end{array}\right.$$

We remark that, by means of this layered approach, we may label a component as faulty even if its probability of being faulty is low, which means that it could actually be a false positive. In other words, we may be privileging recall over precision for our fault prediction task.

A possible way to gauge the trade-off between precision and recall could be not just to compare P(C is not faulty|α) and P(C is faulty|α) and pick the event that carries the highest probability, but declare site C faulty if P(C is faulty|α)>0.5+ω, where ω is a positive threshold. In this case, we would declare a site faulty just if its probability of being faulty is high “enough”. In this way, we would increase the likelihood of outputting a module that is actually faulty. In other words, we would increase the precision of our model, at the expenses of recall.

It is important to note that the layered approach increases the overall scalability and decreases the data imbalance due to the fact that at the first layer, the number of items to be considered as potentially faulty goes down to the number of active sites (as opposed to the total number network components), while the second layer algorithm will be activated only when triggered by the site-level fault prediction.

Finally, another advantage of this approach is the flexibility in the choice of algorithms at each of the layers. Namely, some of the algorithms are well-fitted to work with component-specific data, however, they miss the potential inter-component dependencies among modules on the same site. Then, despite this potential loss of information, the layered approach will still predict site-level faults by using all the information available. We will further elaborate this point in Section 6 dedicated to algorithms.

Naturally, such probability distributions are not given: we can only sample them from experience. Actually, we want to fit a probability distribution to the observed sample distribution and use the fit distributions to make predictions.

Formally, we will need a model for the probability of a site being faulty, $P(C\;is\;faulty|\alpha ,\theta_{site})$, where $\theta_{site}$ are the parameters that define the probability distribution to be fit, and a model for the probability distribution of a module to be faulty, given that the site is faulty $P(\gamma |\alpha ,\theta_{module},$ C is faulty). In the following sections, it will be made clear how such approximation is achieved.

## 5. Data

This section presents the process of building a labeled dataset for training and evaluating the fault prediction algorithms. The data come from a large network and consists of network-wise alarm events, daily snapshots of sites, the list of deployed components on the site, and the list of components sent for repair. Building a labeled dataset from such heterogeneous inputs requires a data consolidation followed by a procedure for attribution of data samples to normal or faulty behaviors. Finally, we put in place a data summarization method allowing us to obtain reusable alarm patterns comparable across different components and sites. These three steps are described in the next subsections.

### 5.1. Data Consolidation

Alarm logs come from two data collection tools and have heterogeneous formats. The first tool monitors the entire network and collects alarm events with start and end timestamps. The second tool collects daily low-level snapshots of sites. A snapshot contains information about the state of alarms on the given site, as well as the list of currently deployed components. The alarms collected by the two tools contain some redundancies but are complementary to a large extent with a predominance of the information coming from the snapshots. We consolidated these data by building a common alarm model with uniquely defined alarm identifiers and a common time dimension. First, alarm events were translated into daily alarm states; the state of a given alarm on a given day was equal to 1 if the alarm was active at least once during the day and was equal to 0 otherwise.

An alarm *a* is identified by a triplet <ResourceInstance,ResourceType,AlarmType>. The first term, ResourceInstance, identifies the logical entity on which the alarm was raised. Here, a logical entity refers to a physical or virtual network resource such as an antenna, a cell, or a fan. The second term, ResourceType, identifies the type of logical entity. For example, *Cell-1* and *Cell-2* are two resource instances of the same resource type *Cell*. Finally, the third term, AlarmType, encodes the meaning of the alarm, e.g., high temperature, or connection loss. It is provided using numerical codes which are composed of 1 to 3 integers. Given the sensitive nature of the dataset, the examples in this paper will contain only the numerical codes of alarm types.

For each alarm $a_{i}$ identified by the triplet $<resInstance_{i},resType_{i},alarmType_{i}>$, we created a binary time series representing the sequence of its daily states. Depending on the prediction layer, the time series were grouped per site or per component. Examples of a time series for a site and a time series for one of its components are shown in Figure 2. The group of component time series, on the right, is a sub-set of the site-level time series on the left.

Finally, in order to obtain homogeneous data objects, the grouped time series were split into bundles using a sliding window of *N* days. An example of a bundle of the component time series of Figure 2 is shown with a red rectangle on the left side of Figure 3.

### 5.2. Data Labeling

In order to train the fault prediction algorithms, we needed a labeled dataset, i.e., examples of alarm bundles that precede faults and examples of bundles not followed by a fault in the (known) future, i.e., normal behavior bundles. To obtain examples of fault cases, we first selected the components in the repair center that had been diagnosed as featuring a hardware fault. This eliminates all the components where no hardware-related issues were detected despite the fact that the component was replaced and delivered to the repair center. Such cases can correspond to configuration errors or software-related issues that do not require intervention on the hardware. Other items could have been removed as a result of wrong problem detection and must not be labeled as fault examples.

Then, we needed to establish the date of replacement of each faulty component. It was obtained by comparing the lists of components in consecutive snapshots of the site where the faulty component was originally deployed. Given the replacement date of a faulty component, we built a 43-day-long time series composed of 28 days before the replacement, the day of the replacement, and the following 14 days. We labeled as a fault examples of all the bundles appearing strictly before the replacement date. We excluded the replacement date which may contain a mixture of alarm signatures specific to the replacement operation and the fault itself. In all cases, the bundles obtained after the replacement were considered to be normal. We completed the set of normal behavior bundles with samples from non-faulty sites randomly selected in the network. We took a one-month margin period to ensure that these sites did not contain a faulty component that had not yet been replaced.

Finally, note that, for the component-level predictions, only the bundles of the faulty component (preceding the replacement) were taken as fault samples, whereas those associated with the other, healthy components on the same site were considered as normal. In contrast, for site-level model training, the bundles were not defined component-wise and all alarms belonging to a given site were used to build the model.

### 5.3. Data Summarization

Each base station of a radio network can have a different configuration and can contain a variable number and different types of components. The goal of data summarization is to abstract from specific resource naming schemes and site configuration variations while maintaining the relevant information for fault prediction. Indeed, alarm identifier triplets <ResourceInstance,ResourceType,AlarmType> as defined in Section 5.1 contain unique resource instance identifiers which make it impossible to compare alarm bundles from different sites or to train a common model.

A simple form of data summarization consists in abstracting from these unique resource identifiers and merging alarm time series that collude on resource type and alarm type but have possibly a different resource instance. If we consider alarm bundles as binary tables with row identifiers defined by the above triplets, then this form of summarization consists in applying a GroupBy operator on the <ResourceType,AlarmType> pair and using an aggregation function such as a logical OR on the corresponding rows of daily alarm states. The resulting *summarized* alarm bundle will now have a standard form defined by the number of unique <ResourceType,AlarmType> pairs as identifiers of the aggregated time series. For example, the time series appearing on the same component with their respective original identifiers <a,x,b> and <a,y,b>, x≠y, will be merged into a single summarized time series with the identifier <a,b> and the time series values equal to 1 whenever either of the original values equals to 1, and 0 - otherwise. Obviously, this operation reduces the original number of time series, In our dataset, we observed a reduction by a factor of 3 on average, see for example Figure 3 in which the left side gives an example of instance time series on one base station and the right side represents the subsequent summarized time series.

This summarization mechanism can be extended to build graph-based summaries instead of aggregated data tables. This can be beneficial because the functional dependencies between resource instances can result in alarm propagation phenomena, which can be better captured by a graph structure. In fact, by analyzing the original raw data bundles one can discover some aspects of these dependency structures.

Let us define an *instance* graph, as G=(N,V), where
(5)$$N=\{n_{i}=<resInstance_{i},resType_{i},alarmType_{i}>,i=1,...,m\}$$
is the set of nodes corresponding to alarm identifier triplets, and
(6)$$V=\{\left(n_{i},n_{j},\rho_{i,j}\right):n_{i},n_{j}\in N,\rho_{i,j}>\epsilon \}$$
is the set of edges, where $\rho_{i,j}$ represents the strength of a dependency relationship between the corresponding alarm time series, which needs to be above some fixed threshold ϵ. Here, one can use a variety of methods to extract such dependencies from binary time series, e.g., Jaccard coefficient [21], cosine measure, or Granger causality [22]. In this work we consider only undirected dependencies, *G* is therefore an undirected graph.

The graph summarization is applied to instance graphs and extends the simple idea of alarm data summarization as follows. The resource instances are removed from alarm identifiers and all nodes with the same resource type and alarm type are merged into a new node. This implicitly merges also the edges of the graph. Note that the instance graph for a given bundle can contain multiple connected components (sub-graphs). In this case, each sub-graph is summarized separately to avoid introducing nonexistent alarm dependency paths in the summarized graph.

More formally, we define an equivalence relation on the set of nodes of the instance graph: $n_{i}∼n_{j}$ if and only if $resType_{i}=resType_{j}$ and $alarmType_{i}=alarmType_{j}$. Let us note ${\bar{n}}_{i}=\left\{n_{j}:n_{j}∼n_{i}\right\}$. Then, the equivalence classes defined by that relation represent the nodes of the summarized graph.

We define $\bar{G}=\left(\bar{N},\bar{V}\right)$, a *summarized* graph, where
(7)$$\bar{N}=\left\{{\bar{n}}_{i}:n_{i}\in N\right\}$$
is the set of nodes corresponding to the type pairs <ResourceType,AlarmType>, and
(8)$$\bar{V}=\left\{\left({\bar{n}}_{i},{\bar{n}}_{j},{\bar{\rho }}_{i,j}\right):{\bar{\rho }}_{i,j}=\max\left(\rho_{h,k}:n_{h}\in \bar{n_{i}},n_{k}\in \bar{n_{j}},\left(n_{h},n_{k},\rho_{h,k}\right)\in V\right)\right\}$$
is the set of edges defined by the maximum dependency value between all pairs of alarm time series belonging to the first and the second equivalence classes respectively. We remark that the choice of the maximum allows convenient propagation of the dependency strength threshold from instance graphs to their summarized representations (e.g., compare with the average or the minimum).

In the sequel, we will use both types of summarization depending on the algorithms used.

## 6. Algorithms

This section presents the algorithms which we developed and/or adapted to perform fault prediction at each level. It first describes the techniques used to identify the faulty sites and then explains how the faulty components were identified on top of site-level predictions.

### 6.1. Site Level Prediction

#### 6.1.1. RNN

Recurrent neural networks (RNNs) are commonly used to deal with sequential data [23]. The RNN architecture is designed to extract the relationships that should hold between the inputs, given their sequential nature. A simple use case is that of predicting the *n*-th element of a sequence, given the first n−1 elements. A concrete example of this type of problem is the task of sentence completion, where the objective of the model is to complete a sentence, given its initial part. In this case, the RNN will output the most likey word, given the observed sequence of preceding words.

In the same fashion as this example, we wanted to train a model that, given a sequence of observed alarms, will output the alarm that is most likely to follow.

The basic idea behind this approach consists in using the trained RNN as an anomaly detector. More precisely, we train the RNN on data coming from logs in normal regime, so that the model will pick up the normal dynamics of alarms by minimizing an error measure between the predicted and the ground-truth alarms. Next, we use the trained model as an anomaly detector. We define it as follows: choose a time-window *t*, a threshold level τ, and an integer number *n*. We calculate the same error measure used to train the model between the predicted alarms and the alarms that are actually registered. If, within a time-window of *t* consecutive days we register at least *n* occurrences of such error being greater that the threshold τ, we declare the site faulty. Refer to Figure 4 for an example of such method.

According to the formalism introduced in Section 3, this would be equivalent to the following formulation. Define the model’s hyperparameters $\theta_{site}=\left(\theta_{RNN},n,\tau ,t\right)$, where $\theta_{RNN}$ are the weights of the RNN. Define $count_{\theta_{site}}\left(C\right)$ as the number of times the error measure exceeds τ for *n* times during a time window of *t* consecutive days for site *C*. Now, the idea is that, if $count_{\theta_{site}}\left(C\right)>0$, then it is likely that *C* is faulty, in other words, that $P\left(C\;is\;faulty|\theta_{site}\right)>P\left(C\;is\;not\;faulty|\theta_{site}\right).$ We use $count_{\theta_{site}}\left(C\right)>0$ as a proxy for $P(C\;is\;faulty|\theta_{site})>0.5+\omega .$

In our dataset, we have 737 distinct summarized alarm identifiers in the form of <ResourceType,AlarmType> pairs, therefore we encode the inputs to the RNN as 737-dimensional binary vectors.

Such vectors represent, for a given day, the alarms that were active. As exemplified in Figure 5, the RNN takes as input a bundle of *n* such vectors, registering alarm activity for *n* consecutive days, and outputs a binary vector to predict the alarm activity at the (n+1)-th day. During training, we use binary cross-entropy between the predicted binary alarm vectors and the true alarm vectors as an error measure to be minimized. This is the same error measure that we used to define the outlier detector. The space of hyperparameters (n,t,τ and the dimension of the RNN hidden state) has been explored and evaluated by cross-validation.

#### 6.1.2. Time-to-Event Analysis

Historically, time-to-event analysis was developed and used by actuaries and medical researchers to measure the lifetime of populations. Their objective was to measure the time duration between birth and death [4]. The main challenge of time-to-event analysis is that, by the end of the study, the event of interest (for example, “death of a patient” in medicine or “failure of an equipment” in reliability engineering) has only occurred for a subset of the observations: the observations may be right-censored. An observation is right-censored if, by the end of the study, the event of interest has not occurred yet for that observation. As mentioned previously, the time duration to the event may be subject to right-censoring; therefore, we need to consider an event indicator in addition to the time duration. In the case when the event did take place, the time duration will indicate the time to the event and the event indicator will be “True” to indicate that the event occurred. In the case when the event has not occurred yet, the time duration will indicate the time to the last reported time and the event indicator will be “False” to indicate that the event has not occurred yet. Since in our settings we have the replacement date (which is a proxy of the failure time), and we have the alarms of all the previous times with their time stamps, the idea of using time-to-event analysis is to make use of this information (i.e., the time duration to the replacement date or the last reported time), something that is missing in classification methods. The basic idea of our approach consists in using a random survival forest (RSF) [24] to predict a risk score for a given day for a given site. As in the previous section, in our dataset we have 737 distinct summarized alarm identifiers <ResourceType,AlarmType>; therefore, we encoded the inputs as 737-dimensional binary vectors. The RSF takes as input the 737-dimensional binary vector for a given day for a given site which corresponds to a bundle of size 1 and outputs a risk score. In time-to-event analysis, we used the information on the event status and the time duration. We added this information to the data: the event indicator and the time duration.

For the normal sites, we know that the site was normal up to the last reported time, therefore

the event indicator will be false for all the observations since there is no event observed in the site;the time duration will be the difference between the last reported time and the observation time.

For the faulty sites, we know that the site was faulty at the replacement date, therefore:the event indicator will be true for all the observations previous to the replacement date;the time duration will be the difference between the replacement date and the observation date;we ignore the remaining observations after the replacement date.

In order to predict whether a site is faulty, we used the RSF method with two modifications: an average rolling window and a risk threshold that will be defined in the following. A random survival forest will predict a risk score for each site for each time-step. For each site, we used an average rolling window (see Figure 6), where the window length *w* was a hyper-parameter to smooth the risk score for a given period. If the average rolling window went above the risk threshold (see Figure 7), where the risk threshold τ was a hyper-parameter, then we classified the site as faulty. Otherwise, we classified it as normal. The space of hyper-parameters (w,τ) has been explored and evaluated by cross-validation.

In the formalism introduced in Section 3, this would be equivalent to the following formulation. Define the model’s hyperparameters $\theta_{site}=\left(\theta_{RSF},w,\tau \right)$, where $\theta_{RSF}$ are the hyperparameters of the random survival forest. Define $\delta_{\theta_{site}}$ as the 0-1 indicator when the average rolling window of length *w* goes above the risk threshold τ. Now, the idea is that, if $\delta_{\theta_{site}}=1$, then it is likely that *C* is faulty, in other words, that $P\left(C\;is\;faulty|\theta_{site}\right)>P\left(C\;is\;not\;faulty|\theta_{site}\right).$ We use $\delta_{\theta_{site}}$ as a proxy for $P(C\;is\;faulty|\theta_{site})>0.5+\omega$.

#### 6.1.3. Classic ML

We compared the RNN and time-to-event analysis at the site level with results achieved by more traditional machine learning (ML) methods used for classification tasks: naive Bayes and decision trees. We used these models to classify alarm bundles as faulty or normal and then aggregate the results of consecutive bundles to determine a prediction for the site. The daily alarms were encoded in a 737-dimension binary vector as described in the previous sections. Different bundle sizes *d* were tested (1<=d<=7). The input for the models was the concatenation of the daily vectors in the bundle, which means that the input for the models would be a binary vector of 737**d* dimensions. Bundles coming from normal sites or from faulty sites after the replacement were labeled as normal and bundles coming from faulty sites before the replacement were labeled as faulty for the training process.

We tested a naive Bayes and a decision tree classifier. The training process was performed following the cross-validation described in Section 7.2. However, the metrics that evaluated the models, in this case, precision and recall, were not computed directly with the results of the models. The reason for this is that we are interested in the capability of the models on classifying at a site level, not in classifying each bundle as faulty or not. To aggregate the results for each site, the labels of consecutive bundles were analyzed and if τ consecutive bundles had a faulty label, then the site was labeled as faulty. Precision and recall were then computed at site-level labels for different hyper-parameter combinations.

### 6.2. Component Level Prediction

This section describes two approaches for identifying the most likely faulty components within the sites declared as faulty in the first layer.

The first approach develops a graph-matching technique using per-component summarized alarm dependency graphs, and the second one builds a random forest classifier for summarized alarm bundles. According to the layered approach, once a site has been classified as containing a fault, we need to solve the second line of Equation (Equation 4):(9)$$\gamma^{*}:=\underset{\gamma \subset C}{arg max}P\left(\gamma |C\;is\;faulty,\alpha \left(\gamma \right)\right).$$
Here, we assume that, in the event of some components γ being faulty, given that the site that hosts them is faulty is independent of the alarms that are emitted from other components, it will be sufficient to condition on α(γ) as opposed to a more general formulation in (Equation 4). This assumption facilitates the design and broadens the choice of component-level prediction algorithms. Furthermore, in practice, we observed that it is extremely rare to have multiple components failing simultaneously. In the sequel, we will assume therefore that γ is a singleton.

#### 6.2.1. Graph Matching Approach

The main idea behind the graph matching-based faulty component identification is to assign a score $r(\gamma_{i})$ to each component $\gamma_{i}\in C$ with respect to the dependencies between alarms $\alpha (\gamma_{i})$ associated with $\gamma_{i}$ and then to select the component with the highest score. Scores can be seen as probabilities after normalization. In the rest of the paper, we will use these scores directly as component fault likelihoods. In order to compute the scores for the components in a given site *C*, the alarms in α are split into a set of bundles $B=\{\alpha \left(\gamma_{1}\right),\ldots ,\alpha \left(\gamma_{n}\right)\}$, one for each component $\gamma_{i}\in C$. The alarm dependency graphs are computed for each bundle using Jaccard correlation with a threshold of 0.6 and transformed into summarized dependency graphs as described in Section 5.3. Note that, due to the thresholding on the dependency significance, each alarm bundle may be represented by a set of connected components (summarized sub-graphs): by abuse of notation, for a summarized graph B∈B we have a set of summarized sub-graphs $B=\{s_{i}\}$, where each sub-graph $\left\{s_{i}\right\}$ can carry a signature of faulty or normal behavior.

The graph matching method has two phases: knowledge model building (training) and faulty component discovery (inference). Due to the layered approach, the knowledge model will be built using only the data from the faulty sites; in fact, it needs to represent the knowledge necessary for distinguishing a faulty component from the others co-located on a faulty site. During the knowledge model building phase, a *contribution score* $g(s_{i})$ is assigned to each summarized sub-graph $s_{i}$ to express the likelihood of a component $\gamma_{j}$ being faulty given the sub-graph $s_{i}$ appears in the set of its summarized sub-graphs $B_{j}$:(10)$$g\left(s_{i}\right)=P\left(\gamma_{j}\;\mathrm{is}\;\mathrm{faulty}|s_{i}\in B_{j}\right)*c\left(s_{i}\right),$$
where the score $g(s_{i})$ is obtained by multiplying the probability $g\left(s_{i}\right)=P\left(\gamma_{j}\;\mathrm{is}\;\mathrm{faulty}|s_{i}\in B_{j}\right)$ by a confidence coefficient $c\left(s_{i}\right)\in \left[0,1\right]$ that reflects the number of times the sub-graph $s_{i}$ was observed within the dataset. The higher the number of observed $s_{i}$, the higher the confidence towards its contributions. Thanks to this mechanism, we avoid rare sub-graphs having a significant (and arbitrary) impact on our fault detection decisions. In this study, we have used the following confidence function:(11)$$c\left(s_{i}\right)=\frac{1}{1+e^{-(\#s_{i}-d)}},$$
where $\#s_{i}$ is the number of occurrences of the sub-graph $s_{i}$ in the training dataset, and *d* is a sensitivity parameter such that for $\#s_{i}=d$ the conditional probability term is attenuated (1+e) times. After optimization of the *d* parameter, its value was fixed at 3 for our tests.

Note that the conditional probabilities, $P(\gamma_{j}\;\mathrm{is}\;\mathrm{faulty}|s_{i}\in B_{j})$, can be statistically derived from the samples of faulty sites in the training dataset. The higher the frequency of a given sub-graph in the faulty samples (and the lower in the normal ones), the higher its contribution score will be. Then, the contribution scores computed according to Equation (Equation 11) allow us to define the knowledge model K in terms of sub-graph and contribution score pairs:(12)$$K=\{\left(k_{i},g\left(k_{i}\right):g\left(k_{i}\right)>\epsilon \right\},$$
where a small threshold ϵ∈[0,1] allows the limiting of the size of the knowledge model by removing the low contribution elements.

The second phase of the graph-matching approach detects faulty components by exploiting the knowledge model K. First, we define a graph-matching score using the Jaccard coefficient applied to the edges of summarized graphs:$$m\left(s_{i},s_{j}\right)=J\left({\bar{V}}_{i},{\bar{V}}_{j}\right)=\frac{|{\bar{V}}_{i}\cap {\bar{V}}_{j}|}{|{\bar{V}}_{i}\cup {\bar{V}}_{j}|}$$
where $s_{i}$ and $s_{j}$ are summarized sub-graphs with their respective sets of edges (without loss of generality, we ignore the edge weights defined in Equation (Equation 8) assuming that a sufficiently selective threshold has been already applied during the construction of instance graphs, Equation (Equation 6)), ${\bar{V}}_{1}$ and ${\bar{V}}_{2}$; the matching score is defined as a ratio between the number of common edges and the total number of distinct edges of the two sub-graphs.

Let a component $\gamma_{i}$ be represented by alarm sub-graphs $B_{i}=\left\{s_{1},\ldots ,s_{n}\right\}$. We would like to determine the score $r(\gamma_{i}|B_{i},K)$ for the component $\gamma_{i}$ being faulty given the (derived) observations $B_{i}$ and the knowledge model K. We first match the sub-graphs from $B_{i}$ with the knowledge model K and extract the list of best matching sub-graphs $K_{i}=\left\{k_{1},\ldots ,k_{n}\right\}$ such that $k_{j}=argmax_{q\in K}\left\{m\left(q,s_{j}\right)\right\}$ for j=1,…,n. In the case of multiple best matches for a given sub-graph $s_{j}$, the one with the highest contribution score is taken. We also store the respective matching scores $m_{j}=max_{q\in K}\left\{m\left(q,s_{j}\right)\right\}$. Now, we have a list of *projected contribution scores*
$\left\{g\left(k_{1}\right)*m_{1},\ldots ,g\left(k_{n}\right)*m_{n}\right\}$ that best characterize the expected impact of the observed sub-graphs in $B_{i}$, and we would like to combine this knowledge to derive a global score $r(\gamma_{i}|B_{i},K)$. To that end, we used the noisy-OR logic that was introduced in [25] to represent multi-causal dependencies even when they have never been observed together. It has been largely used in network diagnosis and fault localization problems [26,27,28] and it allows for compact representation of the impact of multiple independent causes $x_{1},\ldots ,x_{n}$ on the same effect *y* in the form of a posterior probability (all variables being binary):(13)$$P\left(y=1\;|\;x_{1},\ldots ,x_{n}\right)=1-\left(1-p_{0}\right)\prod_{j=1}^{n}{\left(1-p_{j}\right)}^{x_{j}},$$
where $p_{0}$ represents the probability of observing the effect without any cause being active and $\left\{p_{j}\right\}$ are the respective probabilities of each cause to cause the effect. Note that only the terms associated with active causes remain in the product, the inactive cause terms vanish due to taking the power of 0. In our case, we considered the prior probability (or, equivalently, we assumed $p_{0}=Const\neq 0$ for all the components which would not change the outcome of the max-likelihood selection. However, the exploitation of variable prior failure probabilities for different components contains an unexploited opportunity for further improvement) $p_{0}=0$ and $\left\{p_{i}\right\}$ correspond to the projected contribution scores $\left\{g\left(k_{1}\right)*m_{1},\ldots ,g\left(k_{n}\right)*m_{n}\right\}$ of the sub-graphs in $B_{i}$:(14)$$r\left(\gamma_{i}|B_{i},K\right)=1-\prod_{j=1}^{n}\left(1-g\left(k_{j}\right)*m_{j}\right).$$

Finally, the component $\gamma^{*}$ which is most likely to be faulty within a faulty site *C* with respect to the knowledge model K and the observations $\left\{B_{i}\right\}$ is the one with the highest score:(15)$$\gamma^{*}:=\underset{\gamma_{i}\in C}{arg max}r\left(\gamma_{i}|B_{i},K\right).$$
Note that we have the possibility to trade precision against recall by constraining the score of $\gamma^{*}$ to be above a threshold or otherwise assigning γ:=∅. This mechanism has been used in our experiments, reported in Section 7.3.

#### 6.2.2. Random Forest Approach

The main idea behind this approach is to map the second-level faulty component identification to a classic machine learning problem in order to assign a risk score $r(\gamma_{i})$ to each component $\gamma_{i}\in C$. The component with the highest risk score will be predicted to be faulty.

To perform the mapping between the second-level faulty component identification and a classic machine learning problem:We consider as the input features for a random forest classifier the summarized identifier and the alarm presence/absence of 8 consecutive days. A sliding time window will generate different observations. We note that we consider the observation only if the alarm is present at least one day in the considered time window;For the training set, we generate the target as follows: we assign a value of 1 to the input if its summarized identifier is associated to the replaced component, otherwise we assign a value of 0 (see Figure 8);In the test phase, we predict the target (probability of being associated to a faulty component) for each input point. Then, for each component, we recover all the outputs of points that carry the summarized identifiers associated to that component, we sum up all such outputs, and we use this as a risk score for each component. We predict the component with the highest risk score as the faulty component.

## 7. Numerical Results

### 7.1. Data Exploration

Our dataset is composed of a total of 273 alarm logs, in the form of csv files. Such alarm logs are the result of a first calculation where we pre-process the raw data coming from the network logging system. Each alarm log refers to a site, and it records its alarms’ activity during a given time window. For faulty sites, such a time window is set at 43 days; for normal sites, it ranges between 90 and 130 days but most of them are either 110 or 130 days. The time granularity at which alarms are recorded is daily. As already stated, each alarm is identified by the triplet: alarm type, resource type, and resource instance. For each identifier, we have a binary time series recording; for each day, whether this alarm was on (1) or off (0). Out of 273 logs, 73 refer to faulty sites. This means that, for faulty sites, within the 43 days of recorded alarm activity, a fault has occurred at the site.

#### 7.1.1. Number of Alarms

In our dataset, we have a total of 15,990 different full alarm identifiers. Such identifiers are not evenly distributed across sites, we can see in Figure 9 that the number of full alarm identifiers that appear on each site varies from site to site. It is also clear that faulty sites feature a higher number of active alarms each day. This is an indication of the fact that alarms indeed carry useful information to discriminate normal sites from faulty sites.

As described above, our models work with the summarized version of the full alarm identifiers. We have 737 unique summarized alarm identifiers. Figure 10 shows the distribution of the number of single summarized alarms across normal and faulty sites.

#### 7.1.2. Duration of Alarms

The histograms at Figure 11 report the distributions of alarm frequencies. By alarm frequency, we mean the fraction of time during which an alarm is on, with respect to the total duration of time covered by the alarm log. Here, we plot the distribution of alarm frequencies of all logged alarms. We break down the analysis into normal and faulty sites. We see that faulty sites show a heavier right-tailed distribution, indicating that alarms are more active on faulty sites than on normal sites, as it is reasonable to expect.

### 7.2. Nested Cross Validation

In order to evaluate the models, we used the *precision*, *recall*, and $F_{\beta }$ metrics. We remind here that precision = true positives/(true positives + false positives), recall = true positives/(true positives + false negatives), and $F_{\beta }=\left(1+\beta^{2}\right)\frac{precision\cdot recall}{\beta^{2}\cdot precision+recall}$. Since we had a limited number of samples, we resorted to cross-validation to train and test the models. All models for faulty site detection need to be trained in two stages, in a nested cross-validation scheme.

Let us consider the model based on the RNN. First, we needed to train the RNN to predict the alarms occurring at the (d+1)-th day, given the alarms occurring in the previous *d* days. After having performed this on the training set, we needed a validation set containing faulty logs, on which we will gauge the parameters *n* and τ to construct the anomaly detector. Finally, we tested the trained model on a test set that is disjoint from the two sets used during training.

For the survival probability model, we first needed to train the random survival forest model to predict the risk scores for each site for each day. After having performed this on a training set containing normal and faulty sites, we used a validation set to determine the best hyper-parameters of both the random survival forest and the two added hyper-parameters: window (of the average rolling window) and risk threshold.

We split the entire dataset into five subsets and we performed a nested cross-validation. Let us consider the model based on the RNN: the same logic applies to the other models for faulty site-detection. At each round, we picked a subset as a test set and we set it apart. Next, for each possible choice, we picked one of the remaining four subsets as a validation set for the outlier detector parameters, based on the RNN trained on the remaining three subsets. For the outlier detector, we picked the hyper-parameters that, on average across the four inner rounds of cross-validation, yielded the best $F_{\beta }$ scores. We let β vary between 0.1 and 1 with step 0.1, in order to explore the trade-off between precision and recall. Finally, for each choice of such hyper-parameters, we retrained the RNN on all the four subsets constituting the training set and we tested it on the test set. At the end, we averaged out the precision and recall values of the tested models across the five choices of the test set.

Finally, for each choice of such hyper-parameters, we retrained the model on all the four subsets constituting the training set and we tested it on the test set. In the end, we averaged out the precision and recall values of the tested models across the five choices of test set.

With respect to the component prediction layer, the same five-fold split was maintained. For each choice of the test set, the model was trained on the remaining four folds and it was used on top of the site-prediction model, trained on the same four-folds.

### 7.3. Experiments

Here, we report the results of the nested cross-validation procedure explained in Section 7.2.

#### 7.3.1. Site Prediction, RNN Approach

We explored a wide range of hyper-parameter settings. We let the number of neurons in the hidden layer of the RNN vary among the values 50, 100, 200 and 300. We let the time-window parameter *d* vary between 5 and 10. For each choice, we trained the RNN on the training set and, for each trained RNN, we calculated precision and recall on the validation set, letting the threshold parameter τ vary between 0.01 and 0.99 with step 0.01 and letting the window *t* vary from 5 to 15 days and letting the parameter *n* vary between 1 and 5. Out of them, we picked those that maximize the $F_{\beta }$ score for different values of β, and we tested their precision and recall on the test set. In the upper left-side plot of Figure 12, we show the average precision and recall in the validation set and in the final test set. We notice that there is a drop between the validation and the test metrics, suggesting that the model is struggling to generalize to new cases. The exact numerical values are reported in Table 1.

#### 7.3.2. Site Prediction, Survival Probability Approach

We explored the following range of hyper-parameters settings: we let the risk threshold vary between 15 and 30 and we let the average rolling window vary between 1 and 6. For each such choice, we trained the random survival forest on the training set and, for each trained random survival forest, we calculated precision, recall, and $F_{\beta }$ on the validation set. Out of them, we picked those that maximize the $F_{\beta }$ score and we tested their precision and recall on the test set. The results are plotted at the upper right-hand side of Figure 12.

#### 7.3.3. Site Prediction, Classic ML Approach

Similar to the previous methods, the models were tested with different combinations of hyper-parameters and the ones that maximized the $F_{\beta }$ score, for different values of β were selected to compute the precision and recall trade-off on the test set. The average precision and recall in the validation set and in the final test set are reported in Figure 12 and the exact numerical values are reported in Table 2.

From the experiments, it is clear that the survival model based on a random survival forest is the one yielding the best results. Additionally, decision trees show a good performance, whereas naive Bayes show pretty poor results, which may be due to the unbalancing between the two classes in the training data. The method based on RNNs lags behind the models based on decision trees and random survival forests.

#### 7.3.4. Component Prediction

This section presents the results for fault prediction at the component level. Following the layered approach of Section 4, faulty component detection was performed just on sites that have been detected as faulty by one of the site-level algorithms. However, in order to evaluate the performance of component-level algorithms independently from the quality of site-level algorithms, we first present the (conditional) results assuming all faulty sites are correctly identified, i.e., we execute faulty component prediction algorithm just on actually faulty sites, assuming we have an ideal site-level fault prediction with recall = precision = 100%. The accuracy of faulty component identification computed on all faulty sites of the labeled dataset is depicted in Figure 13. It shows the precision-recall trade-off with respect to different score thresholds (in the range [0.5, 1.0]). The higher the score threshold, the higher the precision, and the lower the recall. It also shows that the graph-matching approach largely outperforms the random forest classifier; for this reason, in this section, we skip the further evaluation of the latter method. The numerical values of the precision and recall for this experiment are reported in Table A1 and Table A2 in Appendix A.

The precision-recall trade-off can also be observed when applying faulty component identification on top of the site-level predictions. Figure 14 shows the end-to-end accuracy obtained using the graph-matching approach on top of two hyper-parameter configurations of RNN and survival model-based site-level prediction algorithms. Note that the end-to-end precision and recall are upper bounded by the site-level precision and recall. Additionally, the faulty module identification performs better in terms of precision for the higher precision site-level predictions. For example, when applied on top of the RNN with site-level precision = 100% and recall = 26%; for high score thresholds at the component level, we obtained an end-to-end precision equal to the one of the site-level, i.e., 100%, while for the site level predictions with 92% precision, the best end-to-end precision is getting lower-70%. This means that there are faults that are easier to predict than others at both layers.

Our analysis shows that faults on some components are better predicted. The results per component are sensitive and cannot be presented in detail in this paper. However, we can mention that by selecting the top 15 out of the 18 components present in the dataset, the precision and recall are increased by 20%. The hyperparameters of the RNN models and the survival models used in this experiment are reported in Table 3 and Table 4 in Section 7. The respective numerical values of the precision and recall are reported in Table 5, Table 6, Table 7, Table 8 in Section 7.

## 8. Discussion

We remark that processing raw alarm logs, transforming them into a format compatible with a classification algorithm and defining a reliable procedure to label them was by no means a trivial exercise. The main difficulty resided in integrating different data sources automatically to identify the failed and normal components (labels) and the failure dates. We used the component replacement date as a proxy for the failure date. This inevitably introduced some errors: a faulty hardware piece may require a variable time to reach the repair center; hardware pieces that are not under warranty may not even pass through the repair center. Given the available information, there is little space for improvement in data labeling. A possibility is to exploit data sources that have not been used yet, such as ticket creation dates and work orders that could help us to better identify the failure occurrence date. Another more ambitious and long-term solution would be to radically change the process of hardware repair and substitution, in order to better keep track of all the information that can be useful to automatize the process. For instance, the incorporation of expert feedback, if carried out in a systematic way, can allow us to enrich the labels.

We observed that faulty module prediction relies on the assumption that the failure of a module can be predicted just by looking at its own alarms. This assumption is reflected in conditioning just on α(γ) at Equation (Equation 9). One could argue that the failure of a component could be a consequence of some anomaly of another component, hence this independence assumption may not be verified in practice. Taking into account possible inter-dependencies between modules would naturally lead to modeling the object of interest as a graph, where each node would represent a module. By construction, each base station would be modeled by a different graph. Making predictions on graphs of variable structure poses obvious challenges to most machine learning models. Graph neural networks [29] can be the right tool: this will be a subject of our future research.

## 9. Conclusions

We presented the problem of predicting the failure of wireless network hardware components, leveraging alarm logs produced by the network. In order to overcome the challenges posed by the scarcity of examples of hardware faults, we resorted to a layered approach to fault prediction, where, at the first stage, we pin down the base stations that are most likely to be faulty, and then we applied a second algorithm to detect the faulty modules, restricting ourselves to the components contained in such base-stations.

We explored different algorithms for both tasks. For the first layer, we experimented with approaches based on survival models, RNNs, decision trees, and naive Bayes. On top of these, we experimented with two algorithms for module prediction, namely a graph-matching algorithm and the random forest classifier. We evaluated the performance of algorithms in practical settings on a data set coming from a real radio network deployed by a major telecommunication operator, consisting of more than ten thousand base stations. The alarm logs cover a period of time of roughly one year. From our experiments, the combination of the survival probability model for site prediction and the graph matching algorithm for module prediction yielded the best results.

## Figures and Tables

**Figure 1 entropy-25-00917-f001:**
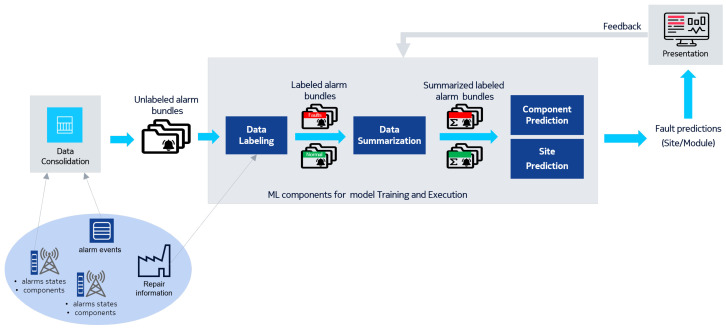
The end-to-end data collection and processing pipeline.

**Figure 2 entropy-25-00917-f002:**
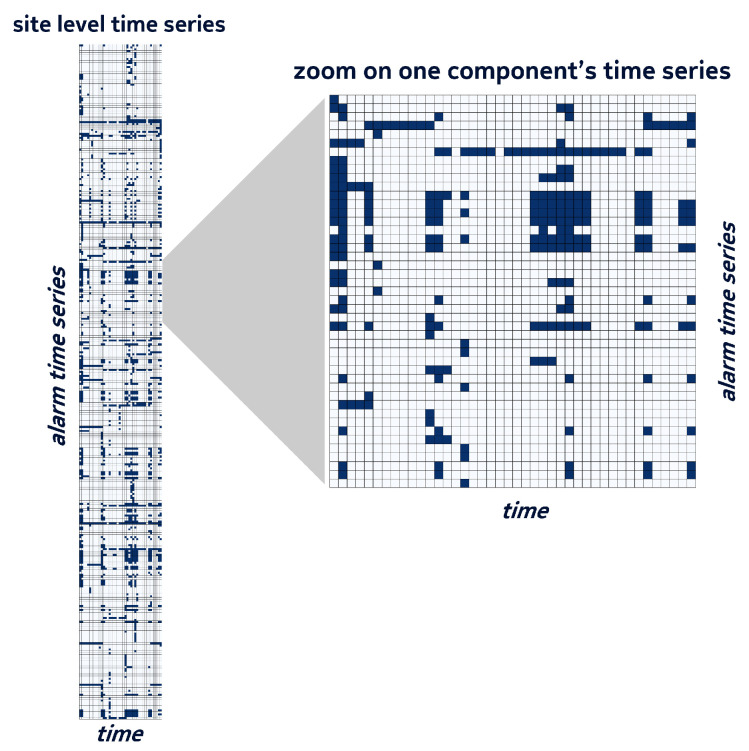
Example of site-level and component-level alarm time series.

**Figure 3 entropy-25-00917-f003:**
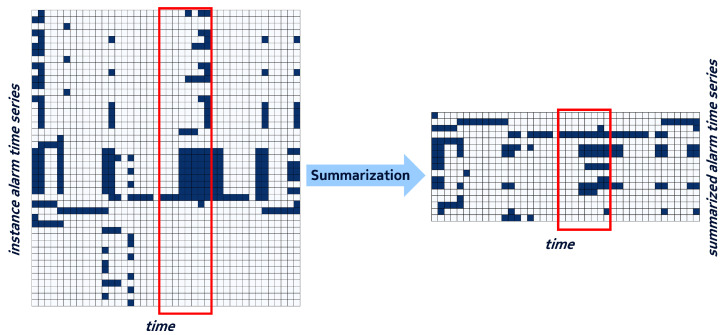
Example of alarm time series before and after summarization. The red boxes delimit an 8-day bundle in their original and summarized forms, respectively.

**Figure 4 entropy-25-00917-f004:**
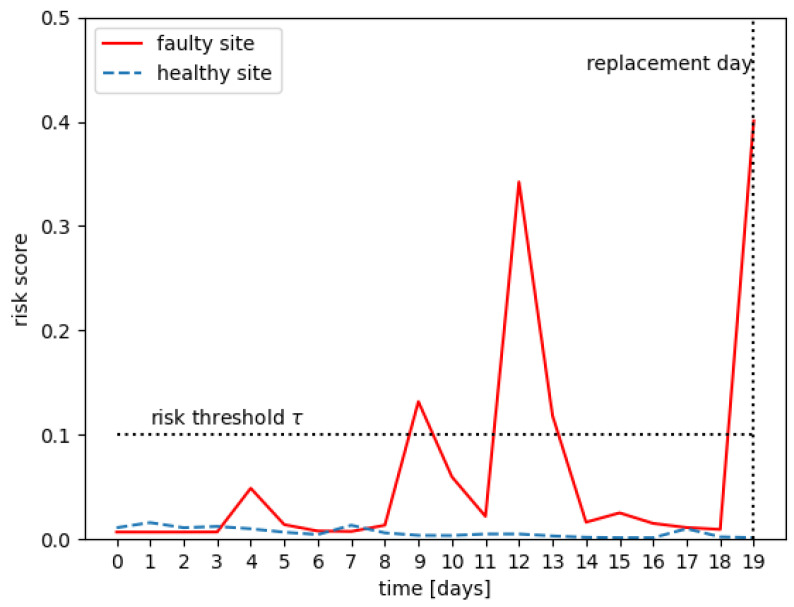
Binary cross-entropy loss for the alarm prediction task for a normal and for a faulty site: the loss on the faulty site is visibly higher than that on the normal site.

**Figure 5 entropy-25-00917-f005:**
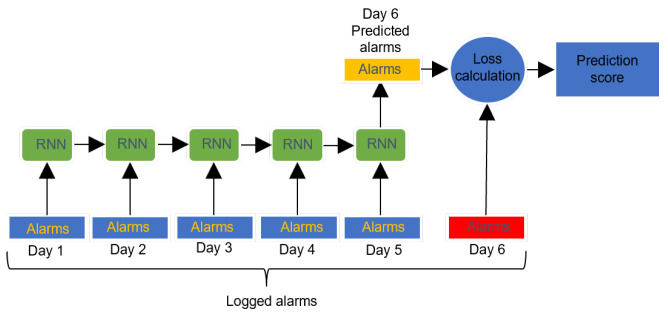
Example of RNN architecture: it processes alarms for five consecutive days and outputs a prediction for the alarms on the sixth day. Such prediction is compared to the alarms actually occurring at the sixth day and a prediction score is calculated leveraging the cross-entropy loss.

**Figure 6 entropy-25-00917-f006:**
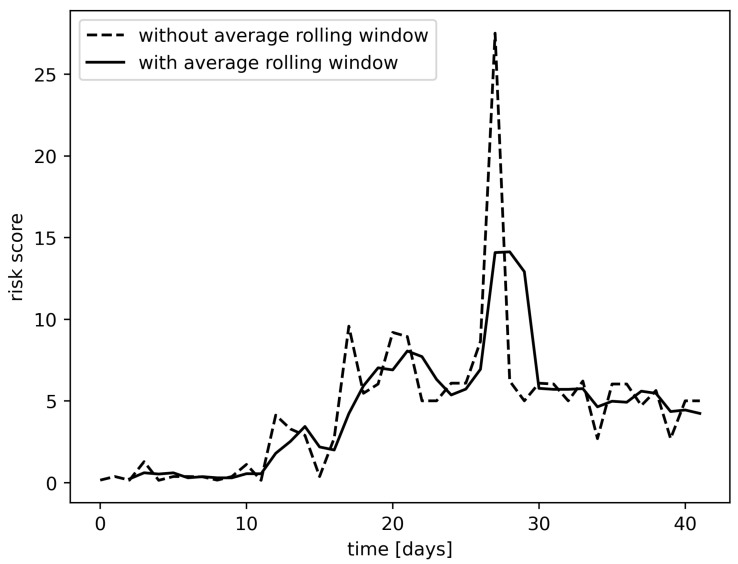
Risk scores without average rolling window vs risk scores with an average rolling window of length 3.

**Figure 7 entropy-25-00917-f007:**
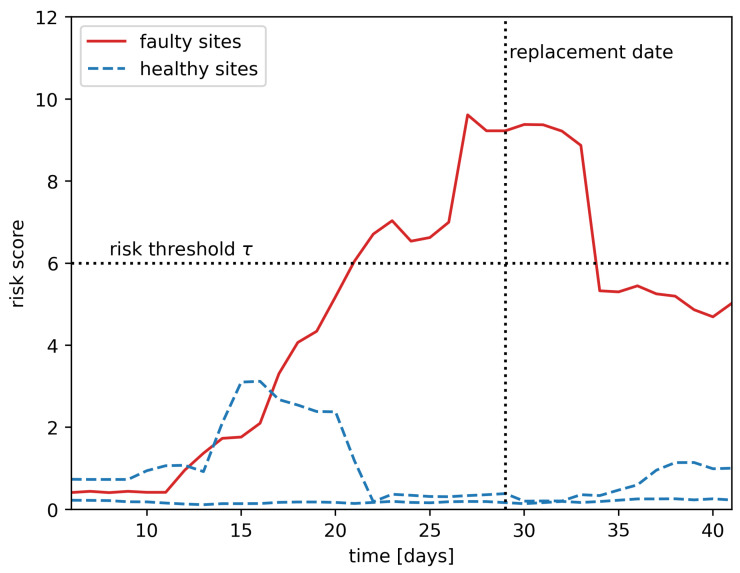
Risk threshold.

**Figure 8 entropy-25-00917-f008:**
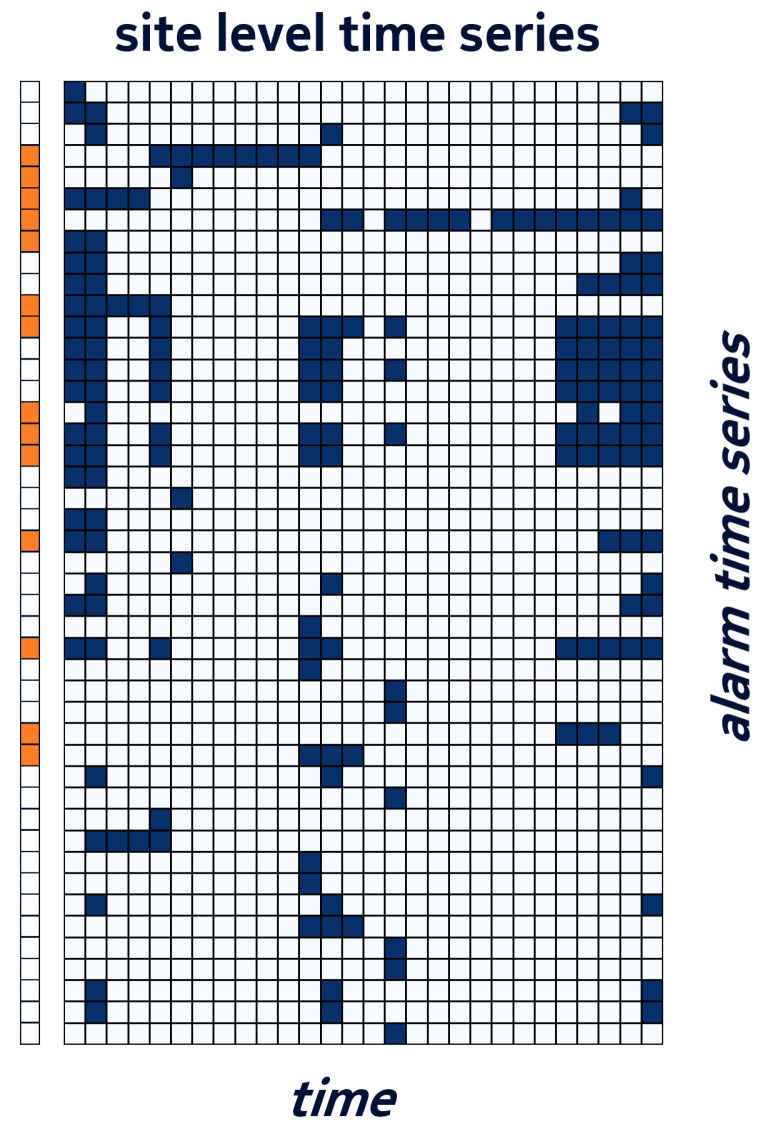
Example of site-level alarms with a left-side column marking the time series of the alarms emitted by the faulty component.

**Figure 9 entropy-25-00917-f009:**
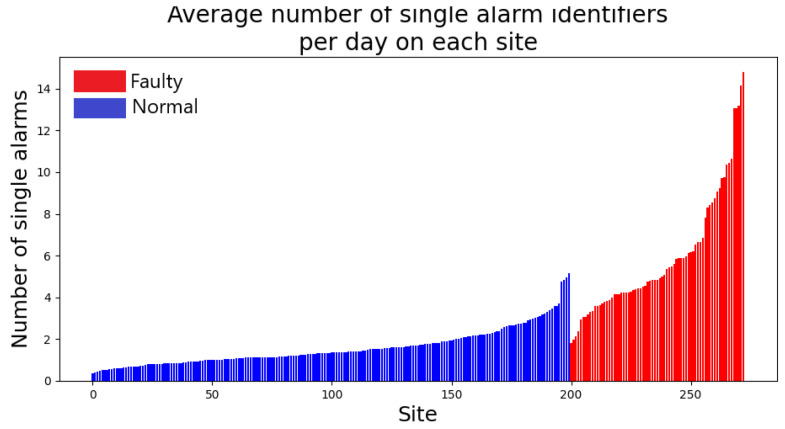
For each site, we count the average daily number of single triplets <ResourceInstance,ResourceType,AlarmType> appearing in its log. We can see that faulty logs feature a higher number of triplets.

**Figure 10 entropy-25-00917-f010:**
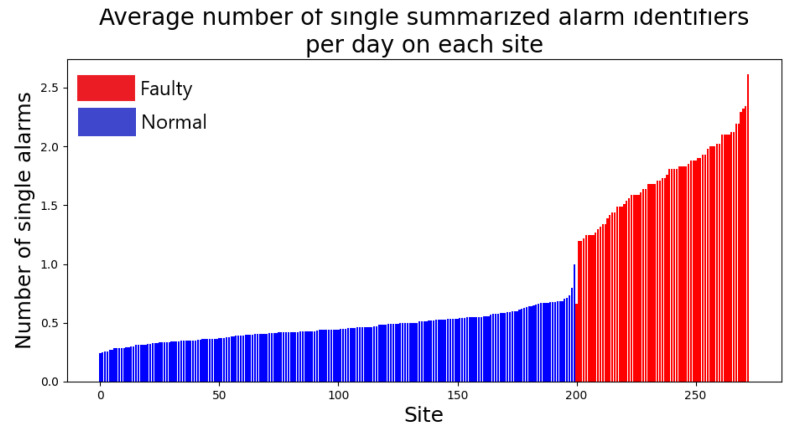
For each site, we count the average daily number of single summarized alarms identifiers <ResourceType,AlarmType> appearing in its log. We can see that faulty logs feature a higher number of summarized alarms identifiers.

**Figure 11 entropy-25-00917-f011:**
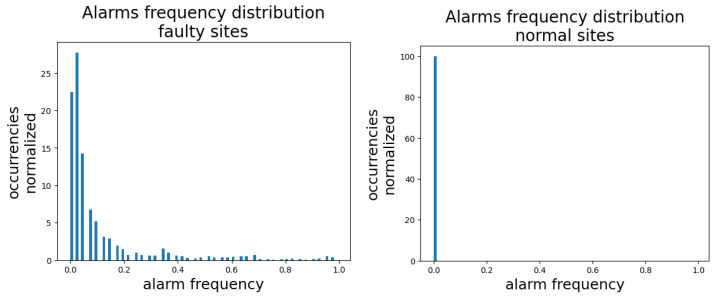
Distribution of alarm frequencies, broken down into normal and faulty sites. For an alarm, its frequency is the fraction of the total observation time during which it has been active.

**Figure 12 entropy-25-00917-f012:**
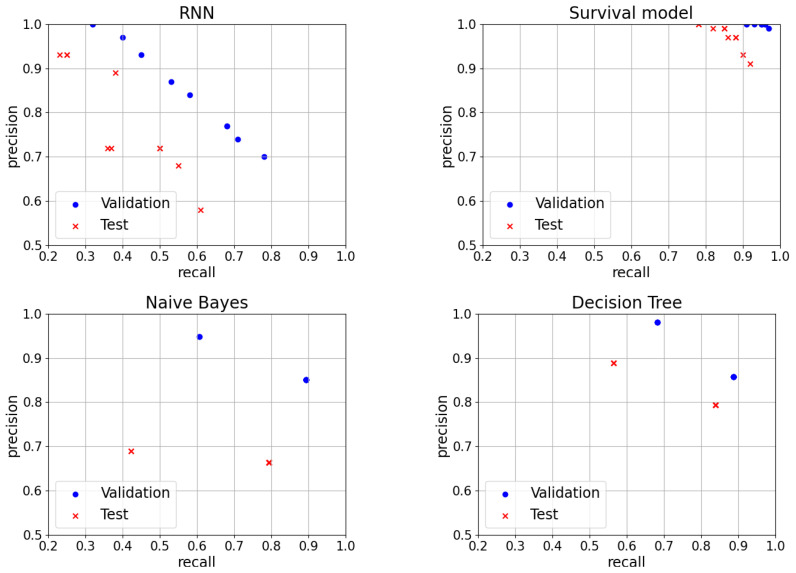
Precision and recall on validation and test set for different values of the β parameter of the four algorithms for faulty site prediction.

**Figure 13 entropy-25-00917-f013:**
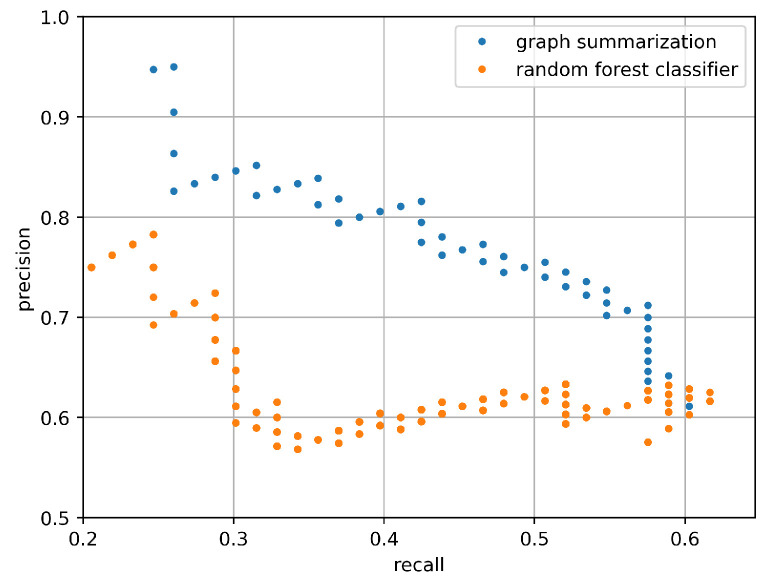
Component level precision-recall for the (summarized) graph matching algorithm and for the random forest classifier, on top of all faulty sites.

**Figure 14 entropy-25-00917-f014:**
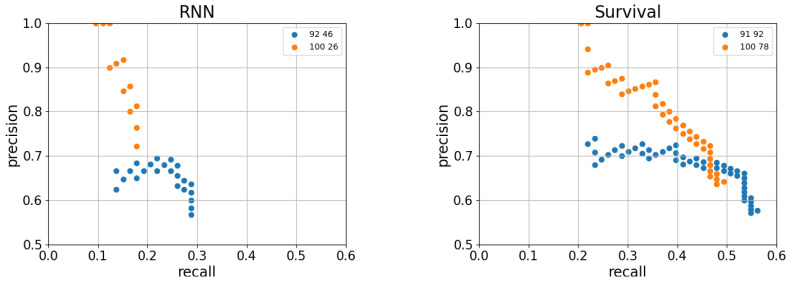
End-to-end precision and recall for the graph-matching module-prediction algorithm, run on top of the RNN-based anomaly detector and the survival probability model for site prediction. For each site-level algorithm, we chose two particular hyper-parameter settings, yielding sufficiently high precision.

**Table 1 entropy-25-00917-t001:** Comparison of prediction scores for site-level fault prediction. RNN anomaly detector and random survival forest model.

	RNN Anomaly Detector				Survival Model			
β	val prec	val Recall	Test prec	Test Recall	val prec	val Recall	Test prec	Test Recall
0.1	1.00	0.32	0.93	0.25	1.00	0.91	1.00	0.78
0.2	1.00	0.32	0.93	0.25	1.00	0.91	1.00	0.78
0.3	0.97	0.40	0.93	0.23	1.00	0.93	0.99	0.82
0.4	0.93	0.45	0.89	0.38	1.00	0.95	0.99	0.85
0.5	0.87	0.53	0.72	0.37	1.00	0.95	0.99	0.85
0.6	0.84	0.58	0.72	0.36	1.00	0.95	0.97	0.86
0.7	0.77	0.68	0.72	0.50	1.00	0.96	0.97	0.88
0.8	0.77	0.68	0.72	0.50	1.00	0.96	0.97	0.88
0.9	0.74	0.71	0.68	0.55	1.00	0.96	0.93	0.90
1	0.70	0.78	0.58	0.61	0.99	0.97	0.91	0.92

**Table 2 entropy-25-00917-t002:** Comparison of prediction scores for site-level fault prediction. Naive Bayes and decision tree models.

	Naive Bayes				Decision Tree			
β	val prec	val Recall	Test prec	Test Recall	val prec	val Recall	Test prec	Test Recall
0.1	0.95	0.60	0.69	0.42	0.98	0.68	0.89	0.56
0.2	0.95	0.60	0.69	0.42	0.98	0.68	0.89	0.56
0.3	0.85	0.89	0.66	0.79	0.98	0.68	0.89	0.56
0.4	0.85	0.89	0.66	0.79	0.98	0.68	0.89	0.56
0.5	0.85	0.89	0.66	0.79	0.86	0.89	0.79	0.84
0.6	0.85	0.89	0.66	0.79	0.86	0.89	0.79	0.84
0.7	0.85	0.89	0.66	0.79	0.86	0.89	0.79	0.84
0.8	0.85	0.89	0.66	0.79	0.86	0.89	0.79	0.84
0.9	0.85	0.89	0.66	0.79	0.86	0.89	0.79	0.84
1	0.85	0.89	0.66	0.79	0.86	0.89	0.79	0.84

**Table 3 entropy-25-00917-t003:** Final RNN models hyperparameters.

RNN Hyperparameter	Model 1	Model 2
Precision-recall	100 26	92 46
hidden layer dimension	50	50
prediction window	9	10
n	2	2
t	18	17
τ	0.27	0.21

**Table 4 entropy-25-00917-t004:** Final random survival forest models hyperparameters.

RSF Hyperparameter	Model 1	Model 2
Precision-recall	100 78	91 92
estimators	100	100
min samples leaf	3	3
window length *w*	3	2
risk threshold τ	19	18

**Table 5 entropy-25-00917-t005:** Precision-recall values for module prediction on top of site prediction performed by the RNN having precision-recall scores of 92–46. For each choice of the score value of the module detection algorithm, the corresponding value of precision and recall is reported.

Score	Precision	Recall
1.0	0.6666666666666666	0.136986301369863
0.999999999972561	0.625	0.136986301369863
0.999999993250194	0.6470588235294118	0.1506849315068493
0.9999999182763856	0.6666666666666666	0.1643835616438356
0.9999996654415968	0.6842105263157895	0.1780821917808219
0.9999993238599096	0.65	0.1780821917808219
0.999998966629584	0.6666666666666666	0.1917808219178082
0.9999970459436154	0.6818181818181818	0.2054794520547945
0.9999967228669364	0.6956521739130435	0.2191780821917808
0.9999806288176908	0.6666666666666666	0.2191780821917808
0.9999497363634728	0.68	0.2328767123287671
0.9994739429566656	0.6923076923076923	0.2465753424657534
0.99872665730684	0.6666666666666666	0.2465753424657534
0.997330480776861	0.6785714285714286	0.2602739726027397
0.9940439557002604	0.6551724137931034	0.2602739726027397
0.9898441806465164	0.6333333333333333	0.2602739726027397
0.958735537828932	0.6451612903225806	0.273972602739726
0.9571723306615252	0.625	0.273972602739726
0.9311094663776294	0.6363636363636364	0.2876712328767123
0.9256615111194504	0.6176470588235294	0.2876712328767123
0.8782285857920709	0.6	0.2876712328767123
0.86200242321312	0.5833333333333334	0.2876712328767123
0.8243559718532313	0.5675675675675675	0.2876712328767123

**Table 6 entropy-25-00917-t006:** Precision-recall values for module prediction on top of site prediction performed by the RNN having precision-recall scores of 100-26. For each choice of the score value of the module detection algorithm, the corresponding value of precision and recall is reported.

Score	Precision	Recall
1.0	1.0	0.0958904109589041
0.999999993250194	1.0	0.1095890410958904
0.9999999182763856	1.0	0.1232876712328767
0.9999993238599096	0.9	0.1232876712328767
0.9999967228669364	0.9090909090909092	0.136986301369863
0.9999497363634728	0.9166666666666666	0.1506849315068493
0.99872665730684	0.8461538461538461	0.1506849315068493
0.997330480776861	0.8571428571428571	0.1643835616438356
0.9940439557002604	0.8	0.1643835616438356
0.9311094663776294	0.8125	0.1780821917808219
0.9256615111194504	0.7647058823529411	0.1780821917808219
0.86200242321312	0.7222222222222222	0.1780821917808219

**Table 7 entropy-25-00917-t007:** Precision-recall values for module prediction on top of site prediction performed by the random survival forest having precision-recall scores of 91-92. For each choice of the score value of the module detection algorithm, the corresponding value of precision and recall is reported.

Score	Precision	Recall
1.0	0.7272727272727273	0.2191780821917808
1.0	0.7391304347826086	0.2328767123287671
0.99999999999983	0.7083333333333334	0.2328767123287671
0.999999999972561	0.68	0.2328767123287671
0.999999999242	0.6923076923076923	0.2465753424657534
0.9999999991643176	0.7037037037037037	0.2602739726027397
0.999999993250194	0.7142857142857143	0.273972602739726
0.9999999926719604	0.7241379310344828	0.2876712328767123
0.9999999719885364	0.7	0.2876712328767123
0.9999999707497808	0.7096774193548387	0.3013698630136986
0.9999999182763856	0.71875	0.3150684931506849
0.9999999002252888	0.7272727272727273	0.3287671232876712
0.9999998119435328	0.7058823529411765	0.3287671232876712
0.9999996654415968	0.7142857142857143	0.3424657534246575
0.9999993238599096	0.6944444444444444	0.3424657534246575
0.999998966629584	0.7027027027027027	0.3561643835616438
0.9999970459436154	0.7105263157894737	0.3698630136986301
0.9999967228669364	0.717948717948718	0.3835616438356164
0.9999920998407088	0.725	0.3972602739726027
0.9999847820736496	0.7073170731707317	0.3972602739726027
0.9999806288176908	0.6904761904761905	0.3972602739726027
0.9999497363634728	0.6976744186046512	0.410958904109589
0.9998125284868672	0.6818181818181818	0.410958904109589
0.9994739429566656	0.6888888888888889	0.4246575342465753
0.9987974751686488	0.6956521739130435	0.4383561643835616
0.99872665730684	0.6808510638297872	0.4383561643835616
0.998562067735804	0.6875	0.4520547945205479
0.9984251938400078	0.673469387755102	0.4520547945205479
0.997330480776861	0.68	0.4657534246575342
0.9964933468118472	0.6862745098039216	0.4794520547945205
0.9940439557002604	0.6730769230769231	0.4794520547945205
0.9911852339664904	0.6792452830188679	0.4931506849315068
0.9898441806465164	0.6666666666666666	0.4931506849315068
0.9728137337618586	0.6727272727272727	0.5068493150684932
0.9691143250982484	0.6607142857142857	0.5068493150684932
0.958735537828932	0.6666666666666666	0.5205479452054794
0.9521379388148312	0.6551724137931034	0.5205479452054794
0.9311094663776294	0.6610169491525424	0.5342465753424658
0.9252920337415372	0.65	0.5342465753424658
0.9218471182472376	0.639344262295082	0.5342465753424658
0.9184575912756672	0.6290322580645161	0.5342465753424658
0.8818724023123204	0.6190476190476191	0.5342465753424658
0.8782285857920709	0.609375	0.5342465753424658
0.86200242321312	0.6	0.5342465753424658
0.8607086329782676	0.6060606060606061	0.547945205479452
0.8487040037363971	0.5970149253731343	0.547945205479452
0.8243559718532313	0.5882352941176471	0.547945205479452
0.7461538461414445	0.5797101449275363	0.547945205479452
0.6842219570583745	0.5714285714285714	0.547945205479452
0.6325107485204375	0.5774647887323944	0.5616438356164384

**Table 8 entropy-25-00917-t008:** Precision-recall values for module prediction on top of site prediction performed by the random survival forest having precision-recall scores of 100-78. For each choice of the score value of the module detection algorithm, the corresponding value of precision and recall is reported.

Score	Precision	Recall
1.0	1.0	0.2054794520547945
1.0	1.0	0.2191780821917808
0.99999999999983	0.9411764705882352	0.2191780821917808
0.999999999972561	0.8888888888888888	0.2191780821917808
0.9999999991643176	0.8947368421052632	0.2328767123287671
0.999999993250194	0.9	0.2465753424657534
0.9999999926719604	0.9047619047619048	0.2602739726027397
0.9999999719885364	0.8636363636363636	0.2602739726027397
0.9999999707497808	0.8695652173913043	0.273972602739726
0.9999999182763856	0.875	0.2876712328767123
0.9999998119435328	0.84	0.2876712328767123
0.9999996654415968	0.8461538461538461	0.3013698630136986
0.999998966629584	0.8518518518518519	0.3150684931506849
0.9999970459436154	0.8571428571428571	0.3287671232876712
0.9999967228669364	0.8620689655172413	0.3424657534246575
0.9999920998407088	0.8666666666666667	0.3561643835616438
0.9999847820736496	0.8387096774193549	0.3561643835616438
0.9999806288176908	0.8125	0.3561643835616438
0.9999497363634728	0.8181818181818182	0.3698630136986301
0.9998125284868672	0.7941176470588235	0.3698630136986301
0.9987974751686488	0.8	0.3835616438356164
0.99872665730684	0.7777777777777778	0.3835616438356164
0.998562067735804	0.7837837837837838	0.3972602739726027
0.9984251938400078	0.7631578947368421	0.3972602739726027
0.9964933468118472	0.7692307692307693	0.410958904109589
0.9940439557002604	0.75	0.410958904109589
0.9911852339664904	0.7560975609756098	0.4246575342465753
0.9898441806465164	0.7380952380952381	0.4246575342465753
0.9728137337618586	0.7441860465116279	0.4383561643835616
0.9691143250982484	0.7272727272727273	0.4383561643835616
0.958735537828932	0.7333333333333333	0.4520547945205479
0.9521379388148312	0.717391304347826	0.4520547945205479
0.9311094663776294	0.723404255319149	0.4657534246575342
0.9218471182472376	0.7083333333333334	0.4657534246575342
0.9184575912756672	0.6938775510204082	0.4657534246575342
0.8818724023123204	0.68	0.4657534246575342
0.8782285857920709	0.6666666666666666	0.4657534246575342
0.86200242321312	0.6538461538461539	0.4657534246575342
0.8607086329782676	0.660377358490566	0.4794520547945205
0.8487040037363971	0.6481481481481481	0.4794520547945205
0.7461538461414445	0.6363636363636364	0.4794520547945205
0.6325107485204375	0.6428571428571429	0.4931506849315068

## Data Availability

Data cannot be publicly shared, since it is proprietary.

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
