# Peer review of "Predicting Network Hardware Faults through Layered Treatment of Alarms Logs"

_entropy, 2023, doi:10.3390/e25060917_

Round 1

Reviewer 1 Report

The authors presented the problem of predicting the failure of wireless network hardware components, leveraging alarm logs produced by the network. Overall, the article is quite interesting. But there are some problems in literature review, description of contribution point, theoretical contribution, innovation and comparison of results. I only agree to the publication of this paper after the authors have revised it according to my comments. Here are some specific suggestions:

1. There are many problems in the grammar of the article, especially in terms of language expression. For example, Maintaining and managing ever more complex telecommunication networks is an increasingly complex task, which often challenges the capabilities of human experts. There are two complexes in this sentence, readability

Difference. In addition, there are many grammatical problems in other parts.

2. Regarding the literature review part of the article, although this article describes the concept of fault prediction methods, it lacks a method description for recent fault prediction tasks, and more recent work needs to be analyzed, such as A variational local weighted deep sub-domain adaptation network for remaining useful life prediction facing cross-domain condition, Remaining useful life prediction of lithium-ion battery with adaptive noise estimation and capacity regeneration detection, An integrated multi-head dual sparse self-attention network for remaining useful life prediction

3. There are too many contribution points in the article, and it is recommended that authors pay more attention to the main contribution of the article. In addition, some contributions, such as the authors customized and applied two families of ML-based methods, Recurrent Neural Nets (RNN) and Survival models achieving high-accuracy results at site-level fault prediction. RNN is a very common method, not enough to contribute

4. The methods used in this article seem to be existing methods. Where are the innovations and theoretical contributions of this article?

5. The comparison of the results of the article is not sufficient. It is only compared with methods such as Naive Bayes and decision tree models. These methods are relatively traditional. It is recommended to use some recent work, such as A parallel hybrid neural network with integration of spatial and temporal features for remaining useful life prediction in prognostics for comparison

6. The conclusion of the article is too long and needs to be streamlined

7. Some pictures in the article are not clear enough, it is recommended to correct the picture format to increase the readability of the article

Please see the comments to the authors.

Author Response

We thank the reviewer for his feedback. Hereafter our replies to each of the points, highlighted in red.

point1

There are many problems in the grammar of the article, especially in terms of language expression. For example, Maintaining and managing ever more complex telecommunication networks is an increasingly complex task, which often challenges the capabilities of human experts. There are two complexes in this sentence, readability 

Difference. In addition, there are many grammatical problems in other parts. 

 reply to point 1

We checked for errors our manuscript. We made an effort to extensively revise the writing, however, in case we are still missing some issues, we would be grateful if the reviewer could pin-point them. 

 point 2

Regarding the literature review part of the article, although this article describes the concept of fault prediction methods, it lacks a method description for recent fault prediction tasks, and more recent work needs to be analyzed, such as A variational local weighted deep sub-domain adaptation network for remaining useful life prediction facing cross-domain condition, Remaining useful life prediction of lithium-ion battery with adaptive noise estimation and capacity regeneration detection, An integrated multi-head dual sparse self-attention network for remaining useful life prediction 

 reply to point 2

The suggested papers have been added to the state of the art, together with some other contributions. The State of the art section was moved at section 2 and provided a more structured overview of it. 

 point 3

There are too many contribution points in the article, and it is recommended that authors pay more attention to the main contribution of the article. In addition, some contributions, such as the authors customized and applied two families of ML-based methods, Recurrent Neural Nets (RNN) and Survival models achieving high-accuracy results at site-level fault prediction. RNN is a very common method, not enough to contribute 

 reply to point 3

As recommended, we focused our contribution to 3 points, the first focusing on the construction of the end to end automation system for predictive hardware mainteinance in large telecommunications networks, the second focusing on the layered approach to fault prediction, and the third focusing on the data summarization procedure and graph matching. The usage of RNNs have been excluded from the contribution points. 

 point 4

The methods used in this article seem to be existing methods. Where are the innovations and theoretical contributions of this article? 

 reply to point 4

As stated in the first point of the contributions, the main contribution of the work is the definition and construction of a completely automated system for predictive hardware maintenance, that goes from raw alarm data collection to the final fault prediction. Other original contributions are the layered scheme to fault prediction, which tackles the problem of class imbalance in the training data, and the data summarization procedure that allows us to use the same ML model on sites having different configurations. 

 point 5

The comparison of the results of the article is not sufficient. It is only compared with methods such as Naive Bayes and decision tree models. These methods are relatively traditional. It is recommended to use some recent work, such as A parallel hybrid neural network with integration of spatial and temporal features for remaining useful life prediction in prognostics for comparison 

 reply to point 5

Precision and recall for the site level prediction using the survival model were very high and already satisfactory, therefore we would not use more complex models when we already saw that a simpler one would yield good results. The reason for the comparison with the basic algorithms is more that of a sanity check. Moreover, the source code for the models that have been suggested to compare our method to are not publicly available, and given the available time to reply to the reviewers’ comments (10 days), it was impossible to implement them from scratch and compare their results to ours. 

 point 6

The conclusion of the article is too long and needs to be streamlined  

reply to point 6

The conclusion and the discussion sections have been separated, and the conclusion is now much shorter and includes the most relevant findings. 

point 7

Some pictures in the article are not clear enough, it is recommended to correct the picture format to increase the readability of the article 

reply to point 7

Figures 2, 3 and 8 were modified to improve readability, if you think any other needs improvement, please let us know. 

Reviewer 2 Report

This paper presents a fault prediction procedure in hardware components in a radio access network using the alarm logs produced by the network elements. The procedure is done in layers, first the detection of the base station faulty is carried out and in a second level the detection of the faulty component of the base station is carried out. In both stages the algorithms used for fault prediction are different.

In the introduction, you must reference the work done for other authors in the same area. Also, the state of the art, i.e., the section 7 of the paper, must be the section 2, to indicate what is new in this paper with respect to the scientific literature in this area.

The sate of the art is very old, you must do a better state of the art, and reference more new papers, papers from the last years, to say that the problem that you are solving is actual and important.

You must explain better the data consolidation, i.e., how do you pass from a alarm a_i to a binary time series. Also, you must explain Figure 2, and you must improve that figure, it is not possible to see anything and to understand that figure.

You must explain better the summarization procedure, and maybe put an example. Also the figure 3 must be improved and better explained.

You must put in the paper what are the hyper-parameters choose for each method, after the cross-validation procedure.

Also, a final table of the results (not only the figures) could be useful to know the utility of the method presented.

Author Response

We thank the reviewer for his feedback. Hereafter, we give our response, in red, to each of the points.

point 1

In the introduction, you must reference the work done for other authors in the same area. Also, the state of the art, i.e., the section 7 of the paper, must be the section 2, to indicate what is new in this paper with respect to the scientific literature in this area. The state of the art is very old, you must do a better state of the art, and reference more new papers, papers from the last years, to say that the problem that you are solving is actual and important. 

reply to point 1

New papers have been added to the state of the art. The state of the art section was moved at section 2 and provided a more structured overview of it. 

Point 2

You must explain better the data consolidation, i.e., how do you pass from a alarm a_i to a binary time series. Also, you must explain Figure 2, and you must improve that figure, it is not possible to see anything and to understand that figure. 

Reply to point 2 

We have improved the write-up for data consolidation, if it still is unclear, please let us know what the unclear points are. The readability of figure 2 has been improved. 

The construction of the time series is now better explained in the paragraph below extracted from the paper. The triplets serve as identifiers of time series, while the values are determined by the state of a given alarm, at a given resource at a given time instant (day): 

“We consolidate these data by building a common alarm model with uniquely defined alarm identifiers and a common time dimension. First, alarm events are translated into daily alarm states; the state of a given alarm on a given day is equal to 1 if the alarm was active at least once during the day and is equal to 0 otherwise.  

An alarm a is identified by a triplet < ResourceInstance, ResourceType, AlarmType >. The first term, ResourceInstance, identifies the logical entity on which the alarm was raised. Here, a logical entity refers to a physical or virtual network resource such as an antenna, a cell, or a fan. The second term, ResourceType, identifies the type of logical entity. For example, Cell-1 and Cell-2 are two resource instances of the same resource type Cell. Finally, the third term, AlarmType, encodes the meaning of the alarm…” 

point 3

You must explain better the summarization procedure, and maybe put an example. Also the figure 3 must be improved and better explained. 

reply to point 3

We clarified the description of the summarization procedure. We also improved the readability of figure 3 and explained it in the text. We have added an example.  

point 4

You must put in the paper what are the hyper-parameters choose for each method, after the cross-validation procedure. 

reply to point 4

In the appendix we included two tables, reporting the hyperparameters of the final models, whose results are reported at figure 13 and 14. 

 point 5

Also, a final table of the results (not only the figures) could be useful to know the utility of the method presented. 

reply to point 5

Such tables have been added in the appendix

Round 2

Reviewer 1 Report

Thanks to the author's revision. I accept its publication.

Thanks to the author's revision. I accept its publication.

Reviewer 2 Report

This paper presents a fault prediction procedure in hardware components in a radio access network using the alarm logs produced by the network elements. The procedure is done in layers, first the detection of the base station faulty is carried out and in a second level the detection of the faulty component of the base station is carried out. In both stages the algorithms used for fault prediction are different.

The authors have answered all my previous questions. So, the paper can be published